# SELF-CONDITIONED DIFFUSION MODEL FOR CONSISTENT HUMAN IMAGE AND VIDEO SYNTHESIS

## ABSTRACT

Consistent human-centric image and video synthesis aims to generate images or videos with new poses while preserving appearance consistency with a given reference image, which is crucial for low-cost visual content creation. Recent advancements based on diffusion models typically rely on separate networks for reference appearance feature extraction and target visual generation, leading to inconsistent domain gaps between references and targets. In this paper, we frame the task as a spatially-conditioned inpainting problem, where the target image is inpainted to maintain appearance consistency with the reference. This approach enables the reference features to guide the generation of pose-compliant targets within a unified denoising network, thereby mitigating domain gaps. Additionally, to better maintain the reference appearance information, we impose a causal feature interaction framework, in which reference features can only query from themselves, while target features can query appearance information from both the reference and the target. To further enhance computational efficiency and flexibility, in practical implementation, we decompose the spatially-conditioned generation process into two stages: reference appearance extraction and conditioned target generation. Both stages share a single denoising network, with interactions restricted to self-attention layers. This proposed method ensures flexible control over the appearance of generated human images and videos. By fine-tuning existing base diffusion models on human video data, our method demonstrates strong generalization to unseen human identities and poses without requiring additional per-instance fine-tuning. Experimental results validate the effectiveness of our approach, showing competitive performance compared to existing methods for consistent human image and video synthesis.

## 1 INTRODUCTION

The field of human-centric image and video generation focuses on creating novel images or videos that conform to specified poses while maintaining appearance consistency with a reference image. Recent advancements Hu et al. (2023); Xu et al. (2023); Wang et al. (2023); Chang et al. (2023) have shown promising human image customizing or animating results, and have potential applications in entertainment, e-commerce, and education. The primary challenge lies in preserving appearance consistency, especially fine details, between the reference image and the generated outputs.

Traditional approaches Chan et al. (2019); Ren et al. (2020); Siarohin et al. (2019); Zhang et al. (2022); Zhao & Zhang (2022); Ren et al. (2022); Han et al. (2018); Yang et al. (2020); Choi et al. (2021); Ge et al. (2021); Xie et al. (2023) typically rely on estimating correspondence between the reference and target images, followed by the use of warping modules to deform the reference image into the target pose. The final results are generated using conditional GANs. However, these methods often struggle to preserve fine details, resulting in artifacts such as low resolution, distortion, loss of detail, and inconsistent appearance, limiting their practical applicability. Recently, diffusion-based models Ho et al. (2020); Saharia et al. (2022); Rombach et al. (2022); Dhariwal & Nichol (2021); Peebles & Xie (2023); Guo et al. (2023); Blattmann et al. (2023a) have shown significant promise in generating photorealistic images and videos. By leveraging these powerful generative models, recent studies Bhunia et al. (2023); Karras et al. (2023); Wang et al. (2023) have produced higher-quality human images and videos compared to GAN-based methods. These models typically

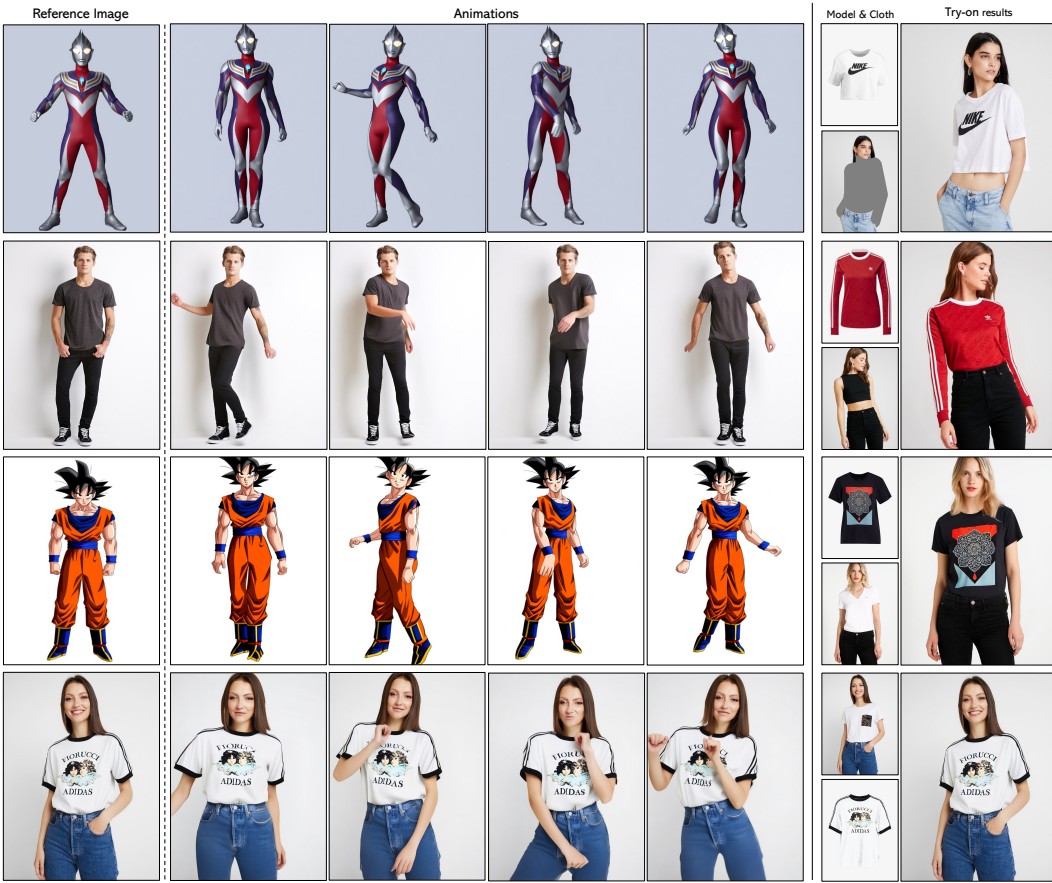

Figure 1: We propose self-conditioned diffusion (SCD) for consistent human image and video synthesis. Left part: our method can generate content-consistent human videos given a reference human image and target poses. Right part: our method can also be applied to visual try-on to maintain the appearance details of the garment.

inject reference image features, extracted by the CLIP image encoder Radford et al. (2021); Karras et al. (2023), into the denoising network or concatenate the reference image with noise along the input channel. However, they still face challenges in preserving fine-grained details: CLIP excels at embedding semantic-level information but struggles to capture discriminative representations necessary to preserve appearance Chen et al. (2023). Similarly, channel concatenation tends to prioritize spatial layout over identity and appearance consistency.

Recent studies Cao et al. (2023); Khachatryan et al. (2023); Zhang et al. (2023) have demonstrated that pretrained text-to-image diffusion models can generate content-consistent images and videos in a zero-shot manner by manipulating the self-attention layers within the denoising network. However, these zero-shot methods often suffer from unstable generation results and struggle to maintain fine-grained details. To address these challenges, AnimateAnyone Hu et al. (2023) and MagicAnimate Xu et al. (2023) introduce an additional trainable copy of the denoising U-Net, known as Reference-Net, to extract appearance features and inject them using the denoising U-Net's self-attention layers during the denoising process. While this approach has set new benchmarks in consistent human image and video generation, these methods typically require substantial resources to train such a large Reference Network. Furthermore, an inconsistent domain gap remains between the reference features extracted by the Reference-Net and the target features in the denoising U-Net, which limits their ability to fully preserve appearance consistency.

In this work, we propose the self-conditioned diffusion (SCD) model for high-quality human-centric image and video synthesis, with a focus on preserving appearance consistency. Unlike previous methods that rely on additional networks to extract reference appearance information, SCD lever-

ages the denoising U-Net itself to directly condition the reference image spatially. This approach ensures that both reference and target features reside within the same feature manifold, enabling better preservation of appearance details compared to Reference-Net-based methods Xu et al. (2023); Chang et al. (2023); Hu et al. (2023). Our approach is inspired by the capability of the pretrained Stable Diffusion model Rombach et al. (2022) to perform zero-shot inpainting and generate harmonious, content-consistent results. By fine-tuning this base model and applying spatial conditioning to the reference image (via spatial concatenation), we effectively preserve both texture and appearance details. To further enhance appearance preservation, we introduce a causal interaction framework within the denoising U-Net, where reference features are restricted to querying from themselves, while target features can query from both reference and target features. This framework ensures that the reference image's fine-grained appearance details are retained throughout the generation process. Furthermore, the spatial conditioning process is decomposed into two sub-processes to allow for a more flexible and efficient generation in the practical implementation: 1) the reference image is passed through the denoising network to extract appearance features, and 2) the target image is then generated by conditioning on these intermediate reference features. As illustrated in Fig. 1, our method synthesizes human images or videos that faithfully maintain the reference appearance while conforming to specified target poses.

Our contributions can be summarized as follows: **1)** We propose a spatial conditioning strategy for reference-based human generation, framing the task as an inpainting problem. The target human image is inpainted under the spatially conditioned reference image, ensuring appearance consistency. **2)** A causal feature interaction mechanism is incorporated within the denoising U-Net to ensure fine-grained preservation of reference appearance details, which allows target features to query from both the reference and target features, while reference features query only from themselves. **3)** We further separate the causal feature interaction framework into two sub-processes: reference feature extraction and subsequent conditioned generation. This design enhances the flexibility, effectiveness, and efficiency of the generation process. **4)** Experimental results demonstrate the effectiveness and competitiveness of our method in generating consistent human-centric images and videos compared to existing methods.

## 2 RELATED WORKS

### 2.1 DIFFUSION MODEL FOR IMAGEN AND VIDEO GENERATION

Diffusion models Sohl-Dickstein et al. (2015); Ho et al. (2020); Song & Ermon (2019); Song et al. (2020b) have significantly advanced visual content generation, achieving superior results and leading the field. In image generation, various diffusion-based methods such as GLIDE Nichol et al. (2021), LDM Rombach et al. (2022), DALLE·2 Ramesh et al. (2022), Imagen Saharia et al. (2022), and DiT Peebles & Xie (2023) have been developed to synthesize photorealistic images that comply with additional class labels or textual descriptions. To enable more controllable synthesis with diffusion models under spatial controls like edge, pose, depth, and segmentation maps, research works such as ControlNet Zhang et al. (2023); Zhao et al. (2024) and T2i-Adapter Mou et al. (2024) incorporate additional controls into pretrained diffusion models by integrating trainable networks. Furthermore, these pretrained diffusion models are also employed for image editing. Dreambooth Ruiz et al. (2023) and Textual Inversion Gal et al. (2022) fine-tune the diffusion model parameters and optimize the textual embedding, respectively, to perform subject-driven image editing. Additionally, some tuning-free methods Meng et al. (2021); Hertz et al. (2022); Tumanyan et al. (2023); Cao et al. (2023) control the denoising process to perform editing without any additional fine-tuning. Building on the success of diffusion models in the image generation domain, researchers have also extended these models for spatiotemporal modeling in video generation Ho et al. (2022b;a); Singer et al. (2022); Hong et al. (2022); Blattmann et al. (2023b); Khachatryan et al. (2023); Guo et al. (2023); Blattmann et al. (2023a). These models have also been explored for video editing Wu et al. (2023); Liu et al. (2023); Geyer et al. (2023); Qi et al. (2023); Ceylan et al. (2023); Yang et al. (2023b), achieving considerable success in terms of visual quality and consistency of the edited videos. Building on these successful visual generation models, we explore consistent human image and video synthesis.

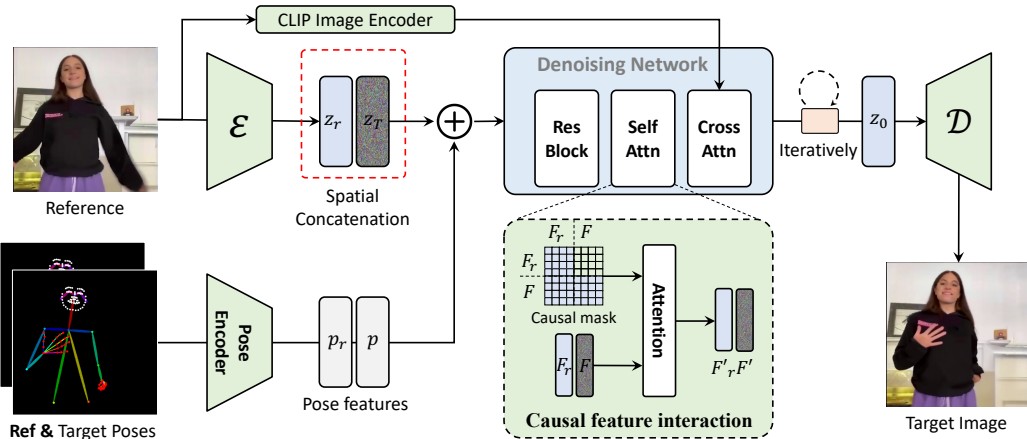

Figure 2: Overview of the self-conditioned diffusion model. Our framework achieves consistent human image and video synthesis by inpainting the desired image under the spatially conditioned reference human image using only the denoising network. A causal feature interaction and reference pose information injection are introduced to further ensure the content consistency between the generated and reference images.

## 2.2 CONSISTENT HUMAN IMAGE AND VIDEO SYNTHESIS WITH DIFFUSION MODELS

Employing diffusion models for synthesizing human-centric visual content has been extensively studied recently, encompassing tasks such as pose transfer Bhunia et al. (2023), human image animation Karras et al. (2023); Hu et al. (2023); Xu et al. (2023), and visual try-on Chen et al. (2023); Huang et al. (2023); Yang et al. (2023a). The primary challenge lies in preserving the texture details and identity of the reference human image. Initial attempts Yang et al. (2023a); Huang et al. (2023); Chen et al. (2023); Wang et al. (2023) aimed to synthesize images with textures similar to the reference image by encoding the reference image using the CLIP image encoder Radford et al. (2021) or the DINO image encoder Caron et al. (2021). However, these methods often struggle to achieve highly detailed texture consistency between the synthesized and reference images. Further research Gou et al. (2023); Bhunia et al. (2023); Kim et al. (2023); Zhu et al. (2023) has explored explicit image warping with flow or implicit warping using attention mechanisms to achieve more consistent edited results. More recently, researchers Hu et al. (2023); Xu et al. (2023); Zhu et al. (2024) have designed Reference-Net-based frameworks that utilize a copy of the denoising U-Net to extract intermediate features and inject them into the denoising U-Net using a reference attention mechanism, thereby achieving much higher consistency in preserving identity and texture details. Meanwhile, this Reference-Net framework has also been adapted to visual try-on Xu et al. (2024); Choi et al. (2024) to achieve better try-on results. In this work, we design a unified diffusion-based framework for human-centric visual content generation and explore self-consistency in the denoising U-Net to achieve appearance consistency between the reference and target images.

## 3 METHOD

### 3.1 SPATIALLY-CONDITIONED DIFFUSION FOR CONSISTENT HUMAN GENERATION

Given a reference human image $I_r$, our objective is to generate new images or videos that preserve the identity of the person in the reference image while adhering to specified target poses. Achieving this objective involves several key challenges: **1)** Maintaining content consistency, including the background, human details, and identity, between the reference image and the generated outputs; **2)** Ensuring that the generated images or videos align accurately with the provided target poses. To address these challenges simultaneously, we propose a self-conditioned diffusion-based model. This model harnesses the denoising network for appearance feature extraction and employs self-conditioning to ensure high-quality, consistent generation of human images and videos.

**Content Consistency through Spatial Conditioning.** Our approach is inspired by the observation that pretrained text-to-image diffusion models, such as Stable Diffusion Rombach et al. (2022), are capable of performing zero-shot inpainting, seamlessly filling masked regions with content that is consistent with the unmasked areas. Additionally, these models can extend a given reference image to a larger one, a process known as outpainting (as shown in Fig. 3(a)). This behavior suggests that pretrained diffusion models inherently generate complete and harmonious images rather than disjoint or fragmented ones. Leveraging this, the added spatial conditioning facilitates the generation of images with a high degree of content consistency.

Motivated by these observations, we propose a spatially-conditioned diffusion model designed for the consistent generation of human images, leveraging large pretrained diffusion models such as Stable Diffusion Rombach et al. (2022). During the training phase, we concatenate the reference image latents with the noisy target image latents along the spatial axis, inputting them into the denoising network. The denoising network predicts the added noise (or other relevant predictions), while the noisy region associated with the target image is cropped to calculate the diffusion loss, as specified in Eq. 4 in the Appendix. As the generation process is conditioned on the reference features extracted by the denoising network itself, we refer to it as self-conditioned diffusion (SCD).

This straightforward spatial conditioning strategy ensures that both reference and target features occupy the same feature domain manifold, thereby enhancing the transfer of appearance details from the reference image to the target. After training the spatially-conditioned model, we can generate content-consistent human images by iteratively denoising the noisy target image with spatial conditioning derived from the reference image. As demonstrated in Fig. 3, this approach effectively produces human images in novel poses while preserving a high degree of consistency in appearance.

**Spatial Conditioning with Causal Feature Interaction.** The spatial conditioning strategy ensures the reference and target features lie within the same feature space, facilitating the transfer of reference appearance details to the target through feature interactions within the denoising U-Net. However, this mutual interaction can potentially compromise the integrity of the reference appearance details. To mitigate this risk, we analyze and implement causal feature interaction within the denoising U-Net to enhance consistency in generation. This approach allows reference features to interact solely with themselves, thereby protecting them from the influence of noisy target features while enabling target images to query appearance information from the reference features. We discuss the internal feature interactions within the spatially-conditioned denoising U-Net.

*How does the spatially-conditioned reference image influence the content of the generated image?* The denoising U-Net of Stable Diffusion comprises multiple basic blocks, each consisting of a residual block He et al. (2016), a self-attention layer, and a cross-attention layer Vaswani et al. (2017). Features from the previous block first pass through the residual block, generating intermediate features. At this stage, feature interactions occur *locally*, particularly in spatially adjacent regions, due to the limited receptive field of the convolution layers. The subsequent self-attention layer facilitates global interactions between the reference and generated image features, allowing the generated image to query comprehensive content information from the reference image through global spatial self-attention. The cross-attention layer, however, only aggregates textual information from the provided textual description to the image features, and thus does not contribute to the interaction between the two types of features. Consequently, with the spatial conditioning strategy, the target image acquires appearance characteristics from the reference image solely through the convolutional and self-attention layers.

*How do these two kinds of modules contribute to content consistency?* To explore this, we conducted an experiment that eliminated interactions with the convolutional and attention layers in our trained spatial-conditioned model, with results illustrated in Fig. 3(b). Using only the convolutional layers for feature interaction can generate a messy image. Using only the convolutional layers for feature interaction resulted in a chaotic image. In contrast, employing solely the self-attention layer yielded generated images that retained a high degree of similarity to the reference image. This disparity can be attributed to the operational differences between the two types of layers: convolutional layers struggle to transfer reference content to the target image in a very localized manner, while self-attention layers can implicitly warp reference features to target features globally.

Based on this analysis, we conclude that self-attention layers play a dominant role in transferring appearance information from the reference to the target images within the spatial conditioning frame-

Masked Image | Inpainted Image | Reference Image | Reposed Image 1 | w/o Conv Interaction | w/o Attn Interaction

(a)  (b)

Figure 3: Examples of content consistency through spatial conditioning. **(a)** Example of zero-shot inpainting with a pretrained Stable Diffusion model Rombach et al. (2022). **(b)** Results of the spatially-conditioned diffusion model with different configurations. The simple spatial conditioning strategy can generate consistent visual humans, and the attention layers play a key role in achieving such consistency.

work. Therefore, we achieve causal feature interaction by constraining interactions between reference and target features specifically within self-attention layers. More precisely, we implement causal attention within the self-attention layers to enhance the preservation of appearance details from the reference image.

**Reference Pose Information Injection.** Previous methods Zhang et al. (2023); Mou et al. (2024); Zhao et al. (2024) have shown that an additional trainable network can effectively encode pose conditions into a pretrained diffusion model, resulting in high-quality images that adhere to specified poses. Building on these approaches, we use a small, trainable pose encoder to extract pose features and integrate them into the target image features, thus controlling the poses of the generated human images and videos. Furthermore, we inject the reference pose features into the denoising network along with the spatial conditioning strategy. This further ensures that reference and target features reside within the same feature space, enhancing the correspondence between the two sets of features within each self-attention layer. Consequently, this leads to improved accuracy in pose control.

**Practical Implementation.** To implement a model with the causal spatial conditioning strategy, we effectively divide the spatially-conditioned generation into two distinct processes as shown in the Fig. 4: reference feature extraction and target image generation. Initially, the reference image $x^r$ is processed through the denoising network to extract the reference features $\epsilon_\theta(x_t^r, t)$. Subsequently, the target image is generated by conditioning on these extracted reference features. Consequently, the objective can be reformulated from Eq. 4 as follows:

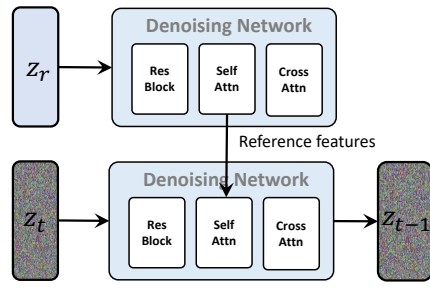

$$\mathcal{L}_{\text{scd}} = \mathbb{E}_{x_0, x_0^r \epsilon, t}(\| \epsilon - \epsilon_\theta(x_t, t, \epsilon_\theta(x_t^r, t, \emptyset)) \|). \quad (1)$$

Figure 4: Separating the causal spatial conditioning process into the reference feature extraction and target image generation.

Note that the $t$ for the reference feature extraction is set to 0 by default. The reference features are injected into the target one with the self-attention layers. Specifically, in $i$-th self-attention layer of the U-Net, the *query* $Q$ is transformed from the target image features, while the *key* $K$ and *value* $V$ features are the concatenations of the reference and target features: $K = [K^r, K], V = [V^r, V]$. Therefore, the target image can effectively query the appearance features by employing the global attention mechanism described in Eq. 5.

## 4 EXPERIMENTS

### 4.1 EXPERIMENTAL SETUP

**Datasets.** In this study, we employ a combination of publicly available and self-collected datasets for training our model. Specifically, for the public datasets, we use the TikTok Jafarian & Park (2021) and UBCFashion Zablotskaia et al. (2019) datasets to train our video model. The TikTok dataset comprises 350 single-person dance videos, each with a duration ranging from 10 to 15 seconds in length. These videos are sourced from TikTok and primarily showcase a human's face and

upper body. The UBCFashion dataset contains 600 fashion videos, of which 500 are for training and 100 for testing. Additionally, we have gathered approximately 3,500 dance videos (about 200 humans) from various online sources to further enhance the generalization capability of our proposed framework. As for the evaluation dataset, to be consistent with previous methods Wang et al. (2023); Xu et al. (2023); Chang et al. (2023); Hu et al. (2023), we utilize 10 TikTok-style videos as the test set for evaluating quantitative metrics.

**Implementation details.** We utilize Stable Diffusion V1.5 Rombach et al. (2022) as our base model for controllable human image generation task, and AnimateDiff Guo et al. (2023) as the video base model for the human animation task. Both models are fine-tuned using the proposed spatial-conditioned diffusion strategy. For training our image model (SCD-I) and video model (SCD-V), we randomly select a single frame from the video to serve as the reference human image. Subsequently, we sample one frame for the SCD-I and 24 frames for the SCD-V as targets. The reference image undergoes a random resized crop, and all frames are adjusted to a resolution of $512 \times 512$. The models are trained using the AdamW optimizer Kingma & Ba (2014) with a learning rate of $1 \times 10^{-5}$ for $30,000$ iterations. The training is conducted on 8 NVIDIA A100 GPUs, employing a batch size of 32 for the SCD-I and 8 for the SCD-V. During the sampling process, we utilize the DDIM sampler Song et al. (2020a) for 25 sampling steps to generate the final outputs. Note that SCD$^\dagger$ is the original straightforward spatial conditioning model, and SCD is spatial conditioning with causal feature interaction.

**Evaluation metrics.** In alignment with previous methods Wang et al. (2023); Choi et al. (2021), we employ several image metrics to evaluate the quality of single images, including FID Heusel et al. (2017), SSIM Wang et al. (2004), PSNR Hore & Ziou (2010), and LPIPS Zhang et al. (2018). Additionally, to evaluate the quality of the animated human video, we report video-level metrics such as FID-VID Balaji et al. (2019) and FVD Unterthiner et al. (2018). In addition to these quantitative evaluations, we also present the generated human images and videos for a qualitative comparison.

## 4.2 COMPARISON TO STATE-OF-THE-ART

We compare the proposed method to the state-of-the-art human animation methods, including **(1)** GAN-based methods FOMM Siarohin et al. (2019), MRAA Siarohin et al. (2021), and TPS Zhao & Zhang (2022); and **(2)** recently diffusion-based methods DreamPose Karras et al. (2023), DisCo Wang et al. (2023), AnimateAnyone Hu et al. (2023)[1], MagicPose Chang et al. (2023), and MagicAnimate Xu et al. (2023). We use their official source codes to obtain the animation results, and utilize the evaluation script from DisCo Wang et al. (2023) for fair comparisons.

**Quantitative results.** Table 1 presents the quantitative performance of various methods on the Tiktok test datasets. Notably, our proposed method, particularly the video-based model SCD-V, achieves highly competitive results. It surpasses state-of-the-art methods such as AnimateAnyone Hu et al. (2023), MagicAnimate Xu et al. (2023), MagicPose Chang et al. (2023), and DisCo Wang et al. (2023) in terms of reconstruction metrics (SSIM, PSNR) and fidelity metrics (LPIPS, FID-VID, FVD). Additionally, our image model SCD-I, also outperforms most existing methods across both image and video metrics. This highly competitive performance can be attributed to the spatially conditioned strategy, which effectively preserves the appearance details of the reference human image. Compared to the Reference-Net-based methods (*i.e.*, MagicAnimate, AnimateAnyone), our methods still demonstrate improvements in most evaluation metrics. These quantitative results underscore the effectiveness and competitiveness of our proposed method in maintaining appearance and identity.

**Qualitative results.** Figure 5 illustrates the qualitative results of various methods on the Tiktok test set. It is worth noting that the large motions in the dance videos and their length pose a significant challenge in preserving the appearance and identity of the reference human image. Existing methods often produce fragmented results or generate frames that do not align with the given pose, especially when the target pose significantly deviates from the reference human's pose. In contrast, our method successfully generates unseen parts (*e.g.*, the hands in the first row of Fig. 5) and highly-detailed frames even under challenging poses (as shown in the third row of Fig. 5). Importantly, our approach achieves this while better preserving the appearance and identity of the reference hu-

---

[1]we use the open-sourced implementation to obtain visual results: https://github.com/MooreThreads/Moore-AnimateAnyone

Table 1: Quantitative comparison of the proposed method against the recent state-of-the-art methods DisCo Wang et al. (2023), MagicPose Chang et al. (2023), MagicAnimate Xu et al. (2023) and AnimateAnyone Hu et al. (2023). Methods with * directly use the target image as the guidance for the animation, including more information than the densepose and pose skeleton. When calculating the metrics, we resize the input image/video to a resolution $256 \times 256$, following Disco Wang et al. (2023). Metrics with ↑ indicate that higher values are better, and vice versa.

| Method | Image Mtrics | | | | | Video Mtrics | |
|---|---|---|---|---|---|---|---|
| | FID ↓ | SSIM ↑ | PSNR ↑ | LPIPS ↓ | L1 ↓ | FID-VID ↓ | FVD ↓ |
| FOMM* Siarohin et al. (2019) | 85.03 | 0.648 | 17.26 | 0.335 | 3.61E-04 | 90.09 | 405.22 |
| MRAA* Siarohin et al. (2021) | 54.47 | 0.672 | 18.14 | 0.296 | 3.21E-04 | 66.36 | 284.82 |
| TPS* Zhao & Zhang (2022) | 53.78 | 0.673 | 18.32 | 0.299 | 3.23E-04 | 72.55 | 306.17 |
| DreamPose Karras et al. (2023) | 72.62 | 0.511 | 12.82 | 0.442 | 6.88E-04 | 78.77 | 551.02 |
| Disco Wang et al. (2023) | 28.31 | 0.674 | 16.68 | 0.285 | 3.69E-04 | 55.17 | 267.75 |
| MagicPose Chang et al. (2023) | 26.67 | 0.692 | 17.03 | 0.270 | 3.33E-04 | 61.73 | 230.88 |
| MagicAnimate Xu et al. (2023) | 32.09 | 0.714 | 18.22 | 0.239 | 3.13E-04 | 21.75 | 179.07 |
| AnimateAnyone Hu et al. (2023) | - | 0.718 | - | 0.285 | - | - | 171.9 |
| SCD-I (Ours) | 33.63 | 0.726 | 18.64 | 0.240 | **2.72E-04** | 33.15 | 153.99 |
| SCD-V (Ours) | 34.44 | **0.731** | **18.81** | **0.236** | 2.75E-04 | **15.58** | **136.60** |

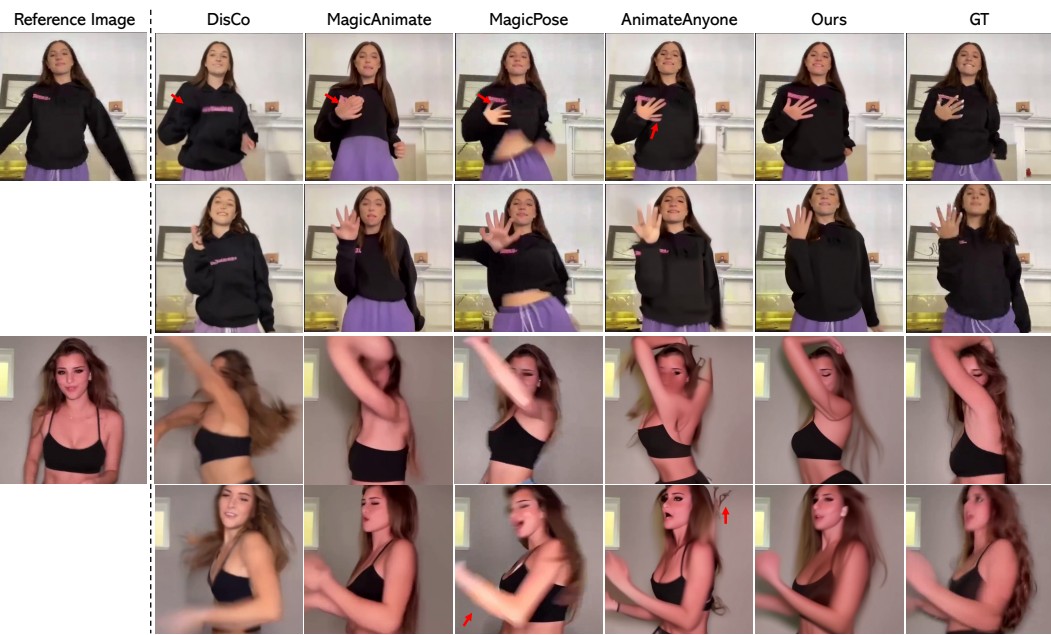

Figure 5: Qualitative comparison results against state-of-the-art human animation methods on the TikTok datasets Wang et al. (2023). The proposed method can generate high-quality human videos complying with the given pose sequence.

man. These highly competitive results further demonstrate the effectiveness of the proposed method, which harnesses the denoising network's capacity to encode appearance information independently, ensuring the reference and target features reside in the same feature space.

### 4.3 ABLATION STUDY

To further validate the effectiveness of our model's design, we conduct ablation studies on our proposed method, focusing on the conditioning strategy and the training data.

**Spatial conditioning strategy.** To evaluate the effectiveness of the proposed spatial conditioning strategy for consistent human image and video generation, we incorporate reference image informa-

Table 2: Ablation results of the proposed method. The best results in each part are bold, and the default setting is grayed. SCD† means the straightforward spatial conditioning strategy without causal feature interaction.

| | FID ↓ | SSIM ↑ | PSNR ↑ | LPIPS ↓ | L1 ↓ | FID-VID ↓ | FVD ↓ |
|---|---|---|---|---|---|---|---|
| CLIP embedding | 79.51 | 0.474 | 12.88 | 0.484 | 6.84E-04 | 106.26 | 690.50 |
| Channel concatenation | 64.60 | 0.577 | 15.09 | 0.390 | 5.15E-04 | 80.92 | 590.36 |
| Reference-Net (w/o SD init) | 48.99 | 0.690 | 16.87 | 0.280 | 3.65E-04 | 40.60 | 303.63 |
| Reference-Net | 41.55 | 0.720 | 18.18 | 0.247 | 3.02E-04 | 32.44 | 172.48 |
| SCD-I† (w/o ref pose) | 34.82 | 0.721 | 18.14 | 0.249 | 3.13E-04 | 36.49 | 218.93 |
| SCD-I† | 33.13 | 0.728 | 18.59 | 0.242 | 2.77E-04 | 32.43 | 164.83 |
| SCD-I (w/o ref pose) | 35.84 | 0.728 | 18.58 | 0.242 | 2.70E-04 | 32.79 | 162.75 |
| SCD-I | 33.63 | 0.726 | 18.64 | 0.240 | 2.72E-04 | 33.15 | 153.99 |

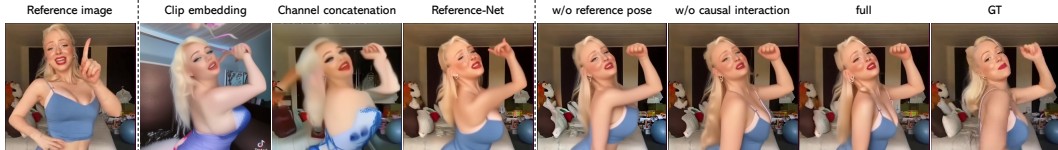

Figure 6: Ablation of different reference image conditioning methods and the key components of the proposed method.

tion using the following alternatives: **1)** CLIP Radford et al. (2021) embedding conditioning, which employs a pretrained CLIP image encoder to extract appearance features and inject them into the denoising U-Net via cross-attention; **2)** Channel concatenation conditioning, which concatenates the reference image latents with the noise map along the channel axis and directly inputs them into the denoising network; **3)** Reference-Net, which utilizes an additional trainable version of the denoising network to extract appearance features, incorporating a reference attention mechanism for injection.

The results are presented in Tab. 2 and Fig. 6. Notably, the CLIP conditioning strategy yields the lowest performance metrics. We hypothesize that, since CLIP is trained on image-text pairs to align the two modalities, it excels at capturing high-level semantic features but struggles to retain the fine-grained appearance details of the reference image. Although both DreamPose Karras et al. (2023) and Disco Wang et al. (2023) employ similar approaches to preserve appearance and identity information, additional strategies are necessary to maintain finer details. For instance, Disco employs a disentangled control strategy and is pretrained on a substantial amount of data to ensure background consistency, while DreamPose conducts model fine-tuning for each reference image to achieve consistent generation. Nevertheless, both methods still fail to preserve the fine-grained details of the reference image (as shown in Fig. 5). The channel concatenation strategy outperforms the CLIP conditioning by better preserving the background but struggles to generate humans in new poses, particularly with limited training data. In contrast, the Reference-Net (as used in AnimateAnyone Hu et al. (2023) and MagicAnimate Xu et al. (2023)) achieves significantly higher metrics and improved visual quality compared to the aforementioned strategies, owing to the fine-grained features extracted by the reference network. However, this approach requires training an additional reference feature extractor and aligning the reference features to the target features. Conversely, our proposed spatial conditioning strategy achieves slightly better performance than the Reference-Net by leveraging the denoising network's inherent capabilities for extracting reference appearance information and generating target images.

**Causal Feature Interaction.** By augmenting the original spatial conditioning with causal feature interaction to more effectively preserve reference appearance information, we observe additional improvements in reconstruction quality metrics, surpassing those of existing conditioning methods (CLIP, channel concatenation, and Reference-Net). These results demonstrate that this strategy successfully mitigates perturbations from noisy target features, thereby enhancing the preservation of appearance information from the reference image and yielding higher reconstruction metrics.

Although our practical implementation of the causal feature interaction is similar to the Reference-Net, our spatial conditioning strategy ensures the reference appearance features reside in the same

feature space as the features of generating the target image using the unified denoising network. The unified feature space is important for content consistency. A straightforward example is that when randomly initializing the Reference-Net rather than using original Stable Diffusion weights, the model performance drastically drops, especially the reconstruction metrics, as shown in Tab. 2, since there is a huge burden in aligning the features from the Reference-Net and the denoising network during the training process.

While our practical implementation of causal feature interaction is similar to that of the Reference-Net, our spatial conditioning strategy ensures that the reference appearance features reside in the same feature space within the denoising network. This unified feature space is critical for maintaining content consistency. A clear illustration of this is shown in Tab. 2. When we randomly initialize the Reference-Net instead of using the original Stable Diffusion weights, the model's performance significantly deteriorates, particularly regarding reconstruction metrics. This decline occurs due to the substantial burden of aligning the features from the Reference-Net with those of the denoising network during the training process.

**Reference pose injection.** We also incorporate reference pose information to enhance the alignment between reference and target features, enhancing their correspondence for more precise pose control. As shown in Tab. 2, the absence of reference pose information injection results in a decline in reconstruction metrics, such as PSNR and SSIM. Furthermore, as illustrated in Fig. 6, without reference pose, the model may produce incorrect target human images. In contrast, injecting reference pose information along with the reference image into the denoising network further ensures the reference and target features reside in the same feature space. Consequently, this strategy further facilitates the generation of target images that better comply with the desired pose.

### 4.4 DISCUSSION AND LIMITATIONS

**More Applications.** Our proposed spatial conditioning strategy can also be effectively applied to other tasks that require appearance preservation, such as visual try-on and face reenactment. For instance, when training our image model SCD-I with the visual try-on dataset VITON-HD Choi et al. (2021), our method facilitates high-quality visual try-on, as illustrated in Fig. 1. The fine-grained textures and text from the garment image can be perfectly transferred to the model, further demonstrating the effectiveness of the proposed self-conditioned strategy. Additional results can be found in the Appendix.

**Limitations.** Despite its effectiveness, our method has certain limitations, particularly in scenarios where the background is complex and the target pose significantly deviates from the reference human image. For instance, during extreme zooming in or out, the appearance and identity may not be perfectly preserved. Additionally, our method struggles to generate perfect faces and hands. We believe that these challenges can be addressed by collecting more high-quality data and employing advanced training strategies. Furthermore, we can apply the spatial conditioning strategy to the more powerful base image and video diffusion models, such as Stable Video Diffusion Blattmann et al. (2023a) and Stable Diffusion 3 Esser et al. (2024), to achieve improved performance.

## 5 CONCLUSION

In this paper, we explore the generation of consistent human-centric visual content through a self-conditioning strategy. We frame consistent reference-based controllable human image and video generation as a spatial inpainting task, in which the desired content is spatially inpainted under the conditioning of a reference human image. Additionally, we propose a causal spatial conditioning strategy that constrains the interaction between reference and target features causally, thereby further preserving the appearance information of the given reference images for enhanced consistency. By leveraging the inherent capabilities of the denoising network for appearance detail extraction and conditioned generation, our approach is both straightforward and effective in maintaining fine-grained appearance details and the identity of the reference human image. Experimental results validate the effectiveness and competitiveness of our method compared to existing approaches. We believe that this self-conditioning strategy holds the potential to establish a new paradigm for reference-based generation.

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

# A    APPENDIX

## A.1    BACKGROUND OF STABLE DIFFUSION

**Diffusion models** Sohl-Dickstein et al. (2015); Ho et al. (2020); Song et al. (2020a); Nichol & Dhariwal (2021) and score-based generative models Song & Ermon (2019); Song et al. (2020b) are a class of probabilistic generative frameworks that learn to reverse the process that gradually degrades the training data distribution. A diffusion model contains a forward and a backward process that adds the noise and removes the noise of the data samples, respectively. During training, the data sample $x_0$ is perturbed to a noisy one $x_t$ by a pre-defined degradation schedule $\alpha_{1:T} \in (0, 1]^T$:

$$q(x_t|x_0) = \mathcal{N}(x_t; \sqrt{\alpha_t}x_0, (1 - \alpha_t)\mathbf{I}); \tag{2}$$

so we can obtain $x_t$ from the clean sample $x_0$ and a Gaussian noise $\epsilon$:

$$x_t = \sqrt{\alpha_t}x_0 + \sqrt{1 - \alpha_t}\epsilon, \tag{3}$$

where $\epsilon \in \mathcal{N}(0, \mathbf{I})$, and $x_T$ coverages to a standard Gaussian for all $x_0$. The reverse process tries to remove the added noise from the noisy sample $x_t$. To achieve this, usually, a denoising network $\epsilon_\theta$ is trained with the objective:

$$\mathcal{L}_{\text{simple}} = \mathbb{E}_{x_0, \epsilon, t}(\| \epsilon - \epsilon_\theta(x_t, t) \|). \tag{4}$$

Once the denoising network has been trained, $x_0$ can be obtained by iteratively performing the denoising process by first sampling $x_T$ from a standard Gaussian. The denoising model is typically realized as a UNet Ronneberger et al. (2015), and the Transformer Vaswani et al. (2017) is further employed currently. Meanwhile, conditioned generation can be achieved when integrating additional conditions like textual description into the denoising model.

**Attention mechanism** is widely integrated into UNet- and Transformer-based diffusion models. Usually, both self-attention and cross-attention are employed in a text-conditioned diffusion model:

$$\text{Attention}(Q, K, V) = \text{Softmax}(\frac{QK^T}{\sqrt{d}})V, \tag{5}$$

where $Q$ is the query feature projected from the noisy image feature, and $K$, $V$ serve as the key and value features projected image features (self-attention) or textual feature (cross-attention). $d$ is the dimension of projected features. With cross-attention layers, textual information can be fused to the generation process, enabling diffusion models to generate images or videos complied with the given textual descriptions Nichol et al. (2021); Rombach et al. (2022). While the self-attention layers try to rearrange the image features, thus they play a crucial role in determining the structure and shape details of the synthesized image Tumanyan et al. (2023). Moreover, in a pre-trained image diffusion, the self-attention layer can be adapted to a crossing one to generate content-consistent images Cao et al. (2023) or temporal-consistent videos Khachatryan et al. (2023).

## A.2    SCD FOR VISUAL TRY-ON

**Datasets and implementation details.**    Our method can also applied to the visual try-on task to generate garment-consistent human images. To validate this, we train our image model SCD-I [2] on the VITON-HD Choi et al. (2021) dataset, which contains 11,647 half-body model images and corresponding garment images at $1024 \times 768$ resolutions for training. Note that we only add noise to the garment region in the human image $x_0$ as input for the denoising network, with the provided garment mark $M$ in the dataset:

$$x_t = (\sqrt{\alpha_t}x_0 + \sqrt{1 - \alpha_t}\epsilon) * M + x_0 + (1 - M). \tag{6}$$

We also apply the garment mask during the loss calculation to ensure the model can inpaint the masked region conditioned on the unmasked human image and the garment image. The model is trained with a learning rate $1 \times 10^{-5}$ for 30,000 iterations, and Adam Kingma & Ba (2014) is employed to optimize model parameters with a batch size of 8.

---

[2]We don't apply the reference pose information injection strategy since the pose of the garment image cannot be extracted.

**Comparison to State-of-the-Art.** We compare the proposed model to the state-of-the-art visual try-on methods, including GAN-based method HR-VITON Rombach et al. (2022), and recently proposed diffusion model-based methods StableVITON Kim et al. (2023), OOTDiffusion Xu et al. (2024), and IDM-VTON Choi et al. (2024). We directly utilize their open-sourced codes to generate the try-on results. The qualitative comparison results are shown in the following figures. We see that our method can generate human images with reference garments and maintain more details of the garments than existing methods. For example, our method can preserve the texts and logos of the reference garments well, while previous reference-net based methods OOTDiffusion and IDM-VTON struggle to achieve this. We attribute the success to our strategy to extract the reference features and synthesize target images using the same denoising network, eliminating the domain gap between the reference and target images.

## A.3 MORE VISUAL RESULTS ON CONSISTENT HUMAN IMAGE AND VIDEO GENERATION

**Failure cases.** As discussed in our main manuscript, our method may fail to generate consistent images and videos in situations where the background is complex and the target pose significantly deviates from the reference human image. As shown in Fig. 9, the complex background cannot be maintained and some artifacts are brought to the foreground human. Furthermore, when the target pose deviates considerably from the reference image, generating the unseen regions becomes challenging, leading to inconsistencies in the appearance of the human subject. To address these issues, we plan to collect high-quality human videos that feature large motions and complex backgrounds. This effort aims to enhance the model's ability to handle diverse scenarios. Additionally, we will explore more advanced base models to improve overall performance and robustness.

**Additional Visual Results.** We present additional animated videos featuring real human subjects and cartoon characters in Fig.11 and the accompanying supplementary video. Our method demonstrates highly competitive performance in animating both real-world individuals and cartoon characters. Furthermore, we can sequentially perform visual try-on and subsequently animate the generated human, as illustrated in Fig.10.

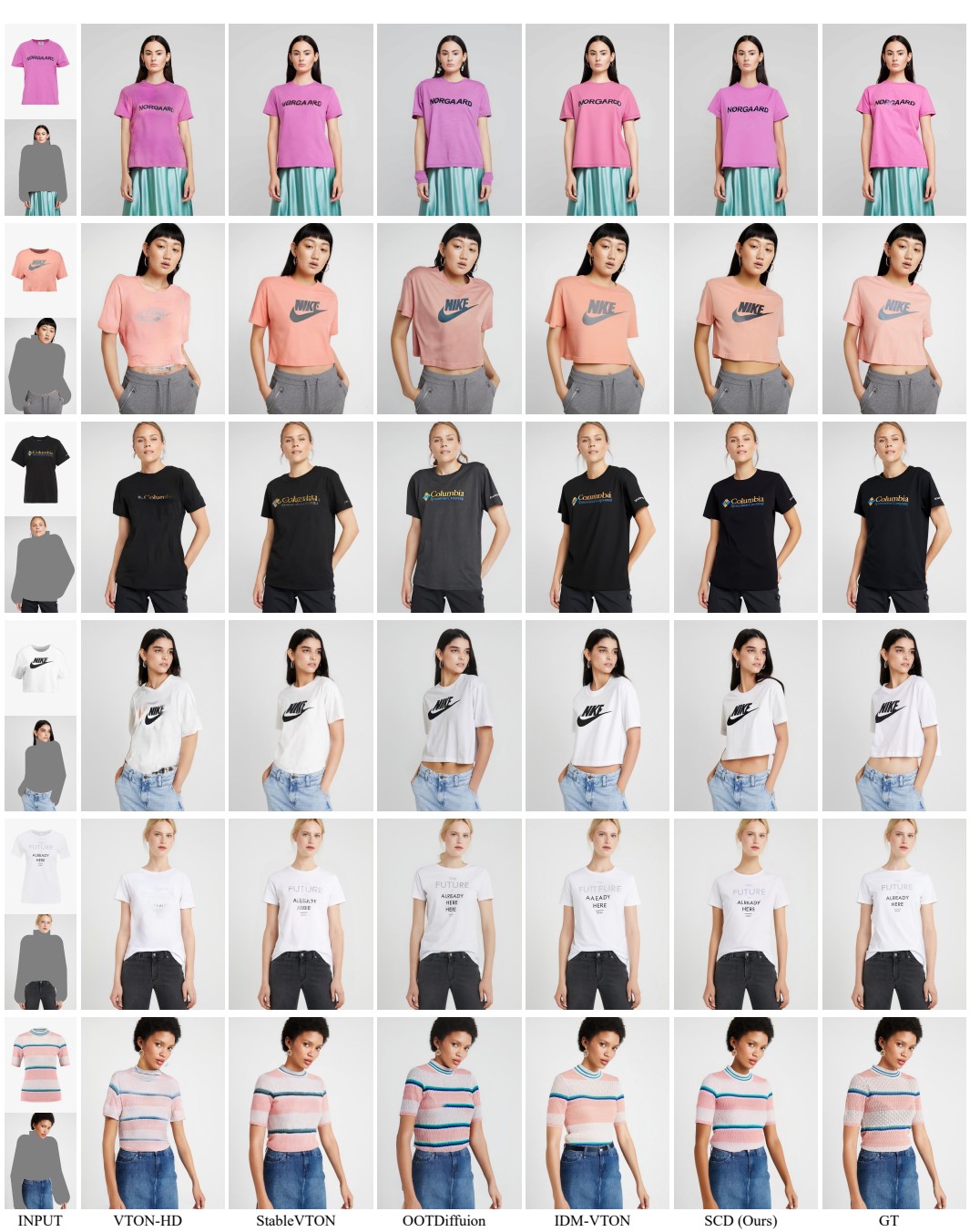

Figure 7: Qualitative comparison on the VITON-HD dataset (paired setting).

INPUT      VTON-HD      StableVTON      OOTD      IDM-VTON      Ours

Figure 8: Qualitative comparison on the VITON-HD dataset (unpaired setting).

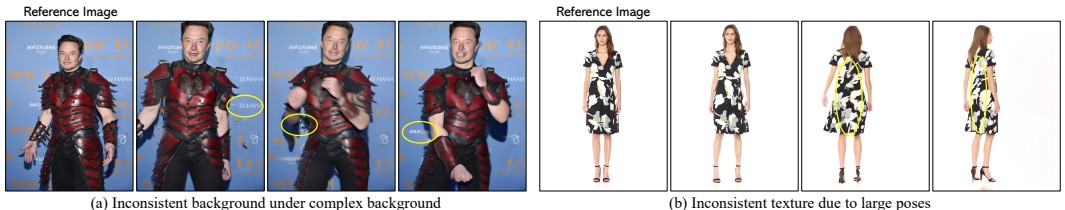

(a) Inconsistent background under complex background      (b) Inconsistent texture due to large poses

Figure 9: Failure cases under complex background and large poses.

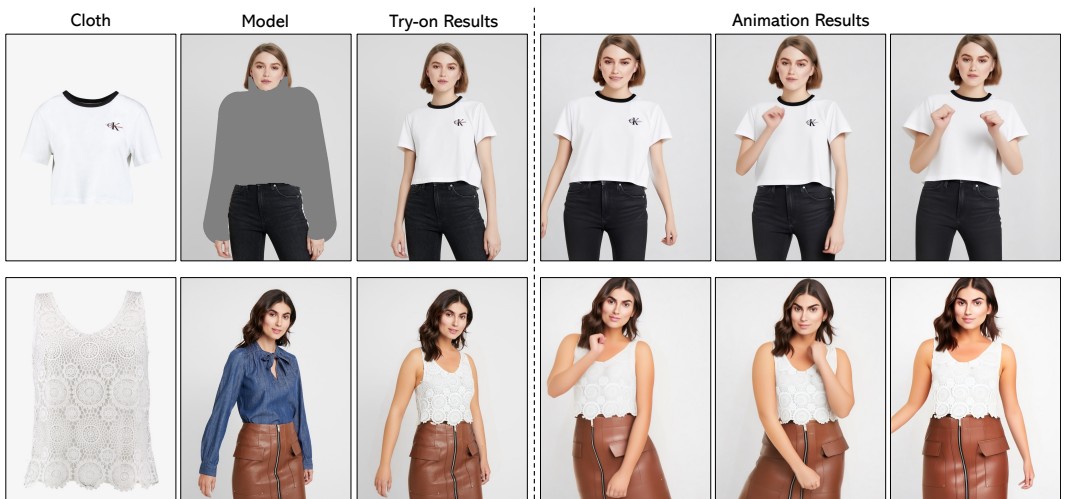

Figure 10: Results of sequentially performing try-on and animation.

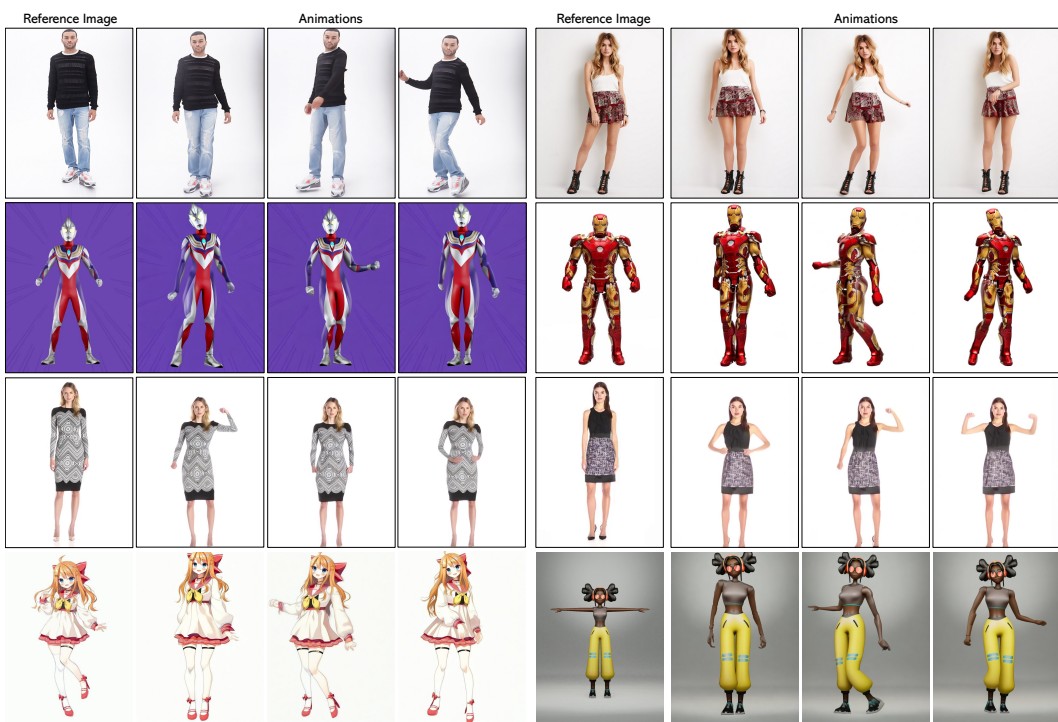

Figure 11: Animation results. Our method demonstrates highly competitive performance in animating real-world or cartoon characters.

