# OpenReview forum: "Self-Conditioned Diffusion Model for Consistent Human Image and Video Synthesis"
_ICLR.cc/2025/Conference — ICLR 2025 Conference Withdrawn Submission_

### Official Review · Reviewer_SNCm · 2024-10-26

**Soundness:** 2
**Presentation:** 3
**Contribution:** 2
**Rating:** 3
**Confidence:** 4

**Summary:**

The paper presents a Self-Conditioned Diffusion (SCD) model designed to enhance consistency in human-centric image and video synthesis. By formulating the task as a spatially conditioned inpainting problem, the model employs a unified denoising network that minimizes domain gaps between reference and target images. The key innovations lie in two aspects: a causal feature interaction mechanism that maintains appearance consistency, and a two-stage generation process that separates reference appearance extraction from conditioned target generation.

**Strengths:**

-  The techniques sound reasonable and the proposed method can enhance content consistency between the generated and reference images.
- The graph is clear and the writing is easy to follow.

**Weaknesses:**

1. **No technical contribution.** The technical novelty is limited. The significance of this paper is not expounded sufficiently. The author needs to highlight this paper’s innovative contributions to prior-guided  I2I/ I2V generation.
2.  **Overclaim and SOTA.** The experimental comparisons are outdated, comparing against older methods while claiming "state-of-the-art" status. The work overlooks recent 2024 publications and lacks quantitative evaluations against recent work. Authors should add necessary discussions and comparisons about some of the following: "Controlnext", "MimicMotion", "Cinemo", "PoseCrafter(ECCV'24)", "Mimo", "X-portrait(SIGGRAPH'24)", "PoseAnimate(IJCAI'24)", "DynamiCrafter(ECCV'24)", "SparseCtrl(ECCV'24)", "LATENTMAN(CVPR'24)" and "Coarse-to-Fine Latent Diffusion for Pose-Guided Person Image Synthesis (CVPR'24)"......

**Questions:**

The paper's qualitative results are inadequate and the image quality is poor. visual examples are insufficient to properly demonstrate the method's effectiveness. I believe there are some issues with the "pose injection" method. The authors should provide more experimental details and show additional generated results.

---

### Official Review · Reviewer_itfT · 2024-10-26

**Soundness:** 3
**Presentation:** 3
**Contribution:** 2
**Rating:** 5
**Confidence:** 5

**Summary:**

This paper proposed a spatial conditioning strategy for human video animation and motion retargeting, building upon the self-attention mechanism proposed in the series of works with Reference-Net. The proposed strategy is efficient and lightweight compared to Reference-Net, and the causal feature interaction mechanism enhances the identity-preserving ability.

**Strengths:**

The paper is well-written and easy to understand. Figure 3 demonstrates the motivation, and Figure 4 clearly explains the details of the proposed causal spatial condition. The proposed pipeline for human motion transfer can be easily extended to virtual try on human image editing, which further shows the effectiveness of the method.

**Weaknesses:**

1. As far as I understand, the proposed method should be quite efficient compared to previous works since there isn't any copied UNet structure. Is there any discussion or comparison of the efficiency, e.g., trainable parameters and inference time for a single batch?

2. What's the difference between the proposed strategy and a "trainable version" of Reference-Only ControlNet, from [here](https://github.com/Mikubill/sd-webui-controlnet/discussions/1236)?  I believe Reference-Only ControlNet also proposed a similar share-weight structure for appearance control. Any detailed discussion on the architecture design?

3. Metric for video generation evaluation. I understand the authors follow previous works and adopt FVD as the video evaluation metric. However, this metric has recently been widely criticized by the community because of its inaccuracy in reflecting the overall quality. I wonder what the performance comparison would be if debiased FVD is used for evaluation. From [here](https://content-debiased-fvd.github.io/)

4. Are there any side-by-side video visualization comparisons between this work and recent baselines? E.g. MagicPose, Champ? It would be better to judge the temporal consistency of the video quality.

5. How does the model generalize to out-of-domain real human identities? E.g. Old people?

6. The denoising network has been fine-tuned on real human datasets and 3500 self-collected dance videos only, but the identity preservation for cartoon-style images in Figure 11 and the supplementary video is quite good. Is there any explanation for this? Do the self-collected videos contain any cartoon characters?

I'm more than **willing** to **raise** my score if my concerns are addressed.

**Questions:**

Please see the weakness.

---

### Official Review · Reviewer_RKw9 · 2024-11-04

**Soundness:** 2
**Presentation:** 2
**Contribution:** 2
**Rating:** 5
**Confidence:** 4

**Summary:**

This paper introduces a human image and video synthesis approach that frames the task as a spatially-conditioned inpainting problem, allowing reference appearance features to guide pose-compliant target generation within a unified denoising network. By using a shared single network with a causal feature interaction framework, the method effectively mitigates domain gaps, enhancing appearance consistency.

**Strengths:**

1. Using the same denoising network for both reference feature extraction and target image generation reduces the training burden and ensures that the target and reference images reside in a consistent feature space.
2. The quantitative results for video synthesis appear promising, demonstrating the SCD-V's effectiveness in maintaining appearance consistency across poses. More video results are preferred if possible.

**Weaknesses:**

1. The logic behind why inpainting is advantageous (Ln221-223) is unclear and requires further clarification. Simply framing the task as inpainting does not inherently address how it enhances appearance consistency.
2. The proposed "causal feature interaction" lacks novelty. It is intuitive that target features should query information from the reference, while reference features should query only from themselves; this approach feels too trivial to be considered a novel contribution.
3. The description of the method in Ln238-287 is overly redundant, especially regarding the use of self-attention in diffusion to achieve content consistency. This observation has already been well-documented in previous video generation research.
4. There are performance concerns. In Table 1, the FID score is significantly higher than other methods, suggesting suboptimal quality. Furthermore, in Table 2, a straightforward spatial conditioning approach without causal feature interaction achieves a lower FID and FID-VID, along with a higher SSIM, which suggests that the main claimed contribution—"causal feature interaction"—does not improve results. In fact, pure spatial conditioning seems sufficient for content consistency. Additionally, Figure 6 shows that results "without causal interaction" are visually closer to the ground truth. Could the authors provide more video-format visual results to clarify?
5. The paper has instances of careless writing (e.g., Ln261) and inconsistencies between titles and tables (e.g., Table 2), which detract from readability and clarity.

**Questions:**

1. Explain more about the intuition from inpainting work.
2. For the performance side, show more results to demonstrate that causal feature interaction does help.

---

### Official Review · Reviewer_Lka8 · 2024-11-04

**Soundness:** 3
**Presentation:** 3
**Contribution:** 2
**Rating:** 5
**Confidence:** 3

**Summary:**

This paper explores controllable human animation generation. Unlike common frameworks that use separate networks for extracting reference appearance features and generating target visuals, this study approaches the task as an inpainting problem. In this framework, the target human image is inpainted based on the spatially conditioned reference image, allowing for the use of a single, unified SD network. Additionally, the paper introduces a causal feature interaction strategy, wherein reference features can only query information from themselves, while target features can access appearance data from both the reference and target images.

**Strengths:**

$\textbf{Originality, Significance}$

1. It is commendable to study controllable human animation generation using a single, unified SD network. This approach may streamline the training process in several areas, such as optimizing GPU resource usage and tuning hyperparameters.

2. The causal feature interaction strategy represents a novel contribution to the single-network paradigm for this task.

3. The method achieves higher scores on the TikTok and UBCFashion datasets compared to previous works.

$\textbf{Clarity}$

1. The paper is easy to follow, and the ideas are well presented.

2. The spatial conditioning and causal feature interaction strategies are validated and discussed in the ablation study section, which is commendable.

**Weaknesses:**

My main concern is that the technical contributions of this paper appear to be incremental.

In the context of controllable human animation generation, I am only knowledgeable about several widely studied works, such as Animate Anyone and Champ. To me, the approach of directly concatenating reference latents and target noise latents spatially for a unified SD diffusion process is new. However, the "inpainting motivation" and spatial conditioning strategy are common in image-to-video generation [1] and multi-view 3D generation tasks [2, 3, 4]. Given that the quantitative improvements are minimal and there are insufficient qualitative comparisons—since the supplemental videos are solely produced by this paper—it is challenging to draw definitive conclusions about the effectiveness of the proposed method.

Regarding the causal feature interaction strategy, it provides only slight improvements, as shown in Table 2 (PSNR: 18.64 vs. 18.59). Based on Figure 6, it seems that the causal feature interaction strategy may not be effective. In fact, it appears that the full model introduces artifacts in the connection region of the shoulder and neck compared to the model that does not utilize the causal feature interaction strategy.


[1] CogVideoX: Text-to-Video Diffusion Models with An Expert Transformer.
[3] One-2-3-45++: Fast Single Image to 3D Objects with Consistent Multi-View Generation and 3D Diffusion. CVPR'24
[4] InstantMesh: Efficient 3D Mesh Generation from a Single Image with Sparse-view Large Reconstruction Models
[5] CRM: Single Image to 3D Textured Mesh with Convolutional Reconstruction Model. ECCV'24

**Questions:**

I have some questions that need further clarification:

1. I noticed that the reported scores in Table 2, such as PSNR, are not consistent with those reported in other works like Champ and MagicAnyone. Could you please clarify this?

2. I am also interested in the experimental settings for using SMPL information as controllable signals within the proposed framework.

---

### Official Review · Reviewer_LPwt · 2024-11-05

**Soundness:** 2
**Presentation:** 3
**Contribution:** 2
**Rating:** 3
**Confidence:** 4

**Summary:**

This paper introduces a self-conditioned diffusion (SCD) model for consistent human-centric image and video synthesis, focusing on maintaining consistency with the reference subject while generating new contents like poses and garments. SCD frames the task as a spatially conditioned inpainting problem, where the reference image as a spatial condition guiding the generation. Besides, the authors introduce a causal feature interaction mechanism to enhances the flexibility and effectiveness. Experimentally, SCD outperforms existing methods in both image and video quality metrics on 10 TikTok-style videos.

**Strengths:**

1. This paper leverages the outpainting ability of the foundation model to complete the generation under the spatial condition of referencing human images through the inpainting, with a novel perspective.
2. The spatial conditions are applied in an inpainting manner, which makes sense.

**Weaknesses:**

1. While the insight of inpainting manner is reasonable, it heavily relies on the capabilities of the foundation model. Since this work takes SD1.5 as the base, which isn’t fully perfect for generating humans, there doesn’t appear to be a mechanism to address the situation when the base model lacks such an ability. This raises reasonable doubt that the effectiveness of the results is largely due to fine-tuning the base model with the dataset rather than overcoming inherent issues. Additionally, base models indeed perform outpainting, but this does not mean their results are consist, there is still a gap.
2. The method primarily focuses on the spatial aspect, with no special treatment on the temporal dimension for video generation—just following the AnimateDiff. So, how to improve the consistency in temporal?
3. The observed phenomenon (line247-267) , whether it is too model-specific (SD) or architecture-specific (UNet-base), this phenomenon may not be universally present. If so, please provide more observation results of models and architectures (DiT), like SD3.5.
4. Dedicating too much of the introduction to detailing previous methods, making it difficult to quickly grasp the main contributions of this paper. It is recommended that the authors reorganize this section, using concise language to summarize the primary limitations of prior work and clearly present contributions of this work.

**Questions:**

Please refer to  the Weakness.

---

### Note · Authors · 2024-11-24

I have read and agree with the venue's withdrawal policy on behalf of myself and my co-authors.